# TRAINING LLMS FOR EHR-BASED REASONING TASKS VIA REINFORCEMENT LEARNING

## ABSTRACT

We present EHRMIND, a practical recipe for adapting large language models (LLMs) to complex clinical reasoning tasks using reinforcement learning with verifiable rewards (RLVR). While RLVR has succeeded in mathematics and coding, its application to healthcare contexts presents unique challenges due to the specialized knowledge and reasoning required for Electronic Health Record (EHR) interpretation. Our pilot study on the MEDCALC benchmark reveals two key failure modes: (1) misapplied knowledge, where models possess relevant medical knowledge but apply it incorrectly, and (2) missing knowledge, where models lack essential domain knowledge. To address these cases, EHRMIND applies a two-stage solution: a lightweight supervised fine-tuning (SFT) warm-up that injects missing domain knowledge, stabilizes subsequent training, and encourages structured, interpretable outputs; followed by RLVR, which reinforces outcome correctness and refines the model's decision-making. We demonstrate the effectiveness of our method across diverse clinical applications, including medical calculations (MEDCALC), patient-trial matching (TREC CLINICAL TRIALS), and disease diagnosis (EHRSHOT). EHRMIND delivers consistent gains in accuracy, interpretability, and cross-task generalization. These findings offer practical guidance for applying RLVR to enhance LLM capabilities in healthcare settings.

## 1 INTRODUCTION

Recent progress in reinforcement learning with verifiable rewards (RLVR) has opened up new opportunities for adapting large language models (LLMs) to complex reasoning tasks (Guo et al., 2025; Jin et al., 2025; Jiang et al., 2025; Lin et al., 2025; Song et al., 2025; Chen et al., 2025; Meng et al., 2025). Rather than relying on dense supervision or handcrafted intermediate annotations, these methods optimize models through outcome-level feedback—rewarding correct final answers while allowing the model to discover its own reasoning path. This makes RLVR a compelling paradigm for tasks where the correctness of the answer can be automatically evaluated using rule-based criteria, but the optimal reasoning process is not explicitly labeled (Xie et al., 2025; Wang et al., 2025; Lyu et al., 2025; Su et al., 2025b; Zhuang et al., 2025; Peng et al., 2025; Luong et al., 2024).

This paradigm is particularly attractive for healthcare applications (Zhang et al., 2025; Kim et al., 2025; Lan et al., 2025; Lai et al., 2025; Qiu et al., 2025; Pan et al., 2025; Su et al., 2025a; Wu et al., 2025). Clinical decision-making often requires multi-step reasoning over noisy Electronic Health Records (EHRs), integrating both structured (e.g., labs, medications) and unstructured (e.g., clinical notes) data (Evans, 2016; Wu et al., 2024). LLMs with reasoning capabilities hold great potential in such settings: they can flexibly process diverse inputs, perform clinical reasoning over extracted variables, and generate interpretable explanations for their predictions (Singhal et al., 2025; 2023; Nori et al., 2023; Li et al., 2024; Wornow et al., 2023b). These capabilities are essential not only for accuracy, but also for transparency and trust, which are key requirements in high-stakes medical domains.

Despite these potentials, RLVR has largely been limited to domains like mathematics and code generation, where pretraining corpora provide substantial coverage (Guo et al., 2025; Team et al., 2025; Yang et al., 2025; Yu et al., 2025; Wu, 2025). In contrast, EHR-based reasoning presents fundamentally different challenges. These tasks involve interpreting noisy clinical data, understanding specialized medical terminology, and performing complex contextual reasoning (Cui et al., 2025;

Jiang et al., 2024; Wang et al., 2024; Lin et al., 2024). These capabilities may not emerge naturally from RLVR training.

To bridge this gap, we introduce EHRMIND, a practical recipe developed through our exploration of RLVR for EHR-based reasoning tasks. Our investigation is guided by two key research questions:

- Q1: Can RLVR training lead to the emergence of medical reasoning capabilities on EHR data?
- Q2: How do supervised fine-tuning (SFT) and RLVR individually and jointly influence model behavior?

We begin with a pilot study on the MEDCALC benchmark (Khandekar et al., 2024), a dataset of medical calculation questions grounded in patient notes. MEDCALC offers a controlled yet realistic testbed, as it (1) assesses core clinical competencies like knowledge, variable extraction, and reasoning; (2) uses clinical notes as input, reflecting real-world practice; (3) enables interpretable evaluation via explicit formulas and rule-based logic; (4) is relatively new, reducing overlap with pretraining corpora.

Surprisingly, we find that even a small 3B LLM (LLaMA-3-3B (Grattafiori et al., 2024)) exhibits strong clinical reasoning capabilities with RLVR alone. However, improvements are not consistent across tasks. To understand this inconsistency (i.e., why RLVR yields large gains in some tasks but limited improvements in others), we examine the base model's behavior and identify two distinct failure modes:

- **Case 1: Knowledge present but misapplied.** The base LLM has relevant medical knowledge but fails to apply it effectively in task-specific clinical contexts. EHRMIND with RLVR alone proves effective here by reinforcing successful reasoning trajectories and helping the model leverage its existing knowledge.
- **Case 2: Knowledge absent.** The base LLM lacks the task-specific domain knowledge. In such cases, reward signals are sparse as the model rarely generates correct answers by chance, making it difficult for RLVR to discover useful updates. To address this, we incorporate a lightweight SFT warm-up phase into EHRMIND. By utilizing a small number of reasoning-annotated examples, we can inject the necessary knowledge and effectively bootstrap RLVR training.

Empirically, we introduce Pass@$k$ as a practical indicator for determining when to apply SFT warm-up. We observe a strong correlation between initial Pass@$k$ on the training set and subsequent RLVR performance gains. Specifically, higher Pass@$k$ values typically correspond to **Case 1 (knowledge present but misapplied)**, where the model possesses task-relevant knowledge and can benefit significantly from RLVR alone. Conversely, low initial Pass@$k$ often suggests **Case 2 (knowledge absent)**, where the base model struggles to produce correct answers even after multiple attempts, indicating that SFT warm-up is necessary.

We further validate EHRMIND on more challenging tasks: patient-trial matching on the TREC CLINICAL TRIALS dataset (Roberts et al., 2021) and disease diagnosis on the EHRSHOT benchmark (Wornow et al., 2023a). Our approach yields substantial improvements: (1) EHRMIND achieves 30–40 absolute improvements on several tasks; (2) EHRMIND discovers clinically meaningful reasoning paths, enhancing model interpretability; (3) EHRMIND demonstrates robust generalization across clinical tasks.

## 2 METHOD

### 2.1 PROBLEM FORMULATION

We consider a general setup for adapting an LLM to perform EHR-based reasoning tasks. Given a task-specific instruction $i$ and a patient-specific EHR $x$ [1], the LLM generates a reasoning path $z$ and a final answer $\hat{y}$ (e.g., a numeric value or classification label). The LLM follows a conditional generation policy $\pi_\theta(z, \hat{y} \mid i, x)$, parameterized by $\theta$. The objective is to find a policy that maximizes expected task performance:

$$\max_\theta \; \mathbb{E}_{i,x,y \sim P(I,X,Y), \; z,\hat{y} \sim \pi_\theta(Z,\hat{Y}|i,x)} \left[ f(\hat{y}, y) \right], \tag{1}$$

where $P(I, X, Y)$ denotes the empirical distribution over task inputs, and $f(\hat{y}, y)$ evaluates prediction quality (e.g., exact match, accuracy).

---

[1]We convert both structured (e.g., diagnoses, medications, lab results) and unstructured (e.g., clinical notes) EHR data into textual form.

## 2.2 THE EHRMIND FRAMEWORK

We aim to provide a practical recipe to improve the performance of pretrained LLMs on EHR-based reasoning tasks. Starting from a general-purpose LLM (e.g., LLaMA-3 Grattafiori et al. (2024)), EHRMIND offers two training variants: (1) direct RLVR, and (2) RLVR with SFT warm-up. We describe both below and compare them empirically in the next two sections.

### 2.2.1 PURE REINFORCEMENT LEARNING

In this variant, the LLM receives a task instruction $i$ and a patient EHR $x$, and generates a reasoning trace $z$ and a final answer $\hat{y}$. A scalar reward $r = f(\hat{y}, y)$ is then computed against the ground truth. This reward serves as a feedback signal indicating whether the current policy should be reinforced or penalized. Over time, the LLM updates its generation policy $\pi_\theta$ to produce higher-reward responses, thereby improving its expected task performance. We refer to this variant as EHRMIND-RLVR.

Motivated by DeepSeek-R1 (Guo et al., 2025), we adopt a rule-based reward function $f(\hat{y}, y)$ that evaluates the correctness of the final answer $\hat{y}$ (e.g., via exact match or classification accuracy). Compared to neural reward models, rule-based metrics are simple, stable, and immune to reward hacking. They also eliminate the need for training or maintaining an additional reward model, keeping the optimization pipeline lightweight.

To perform reinforcement optimization, we adopt Group Relative Policy Optimization (GRPO) (Shao et al., 2024). Compared with traditional algorithms such as Proximal Policy Optimization (PPO) (Schulman et al., 2017), GRPO is significantly more memory-efficient, as it avoids using a separate critic model and estimates the policy gradient using a group-based baseline. Specifically, for each input query $q = (i, x)$, GRPO samples $G$ responses $\{o_1, o_2, \ldots, o_G\}$ from the old policy $\pi_{\theta_{old}}$, where each response $o_j = (z_j, \hat{y}_j)$ consists of a reasoning path and a final answer. Each response is assigned a scalar reward $r_j = f(\hat{y}_j, y)$. These rewards are then normalized within the group to compute the advantage: $A_j = \frac{r_j - \text{mean}(\{r_1, \ldots, r_G\})}{\text{std}(\{r_1, \ldots, r_G\})}$. Here, the advantage $A_j$ reflects how much better a response is than others in the same group. The policy $\pi_\theta$ is then optimized by maximizing the clipped GRPO objective:

$$\mathcal{J}_{\text{GRPO}}(\theta) = \mathbb{E}_{q \sim P(Q), \{o_j\}_{j=1}^G \sim \pi_{\theta_{old}}(O|q)}$$

$$\left[ \frac{1}{G} \sum_{j=1}^G \left( \min \left( \frac{\pi_\theta(o_j \mid q)}{\pi_{\theta_{old}}(o_j \mid q)} A_j, \ \text{clip} \left( \frac{\pi_\theta(o_j \mid q)}{\pi_{\theta_{old}}(o_j \mid q)}, 1 - \epsilon, 1 + \epsilon \right) A_j \right) - \beta \, \mathcal{D}_{\text{KL}}(\pi_\theta \| \pi_{\text{ref}}) \right) \right], \tag{2}$$

where $\epsilon$ and $\beta$ are hyperparameters, and the KL divergence term is used to constrain the updated policy from drifting too far from a reference model $\pi_{\text{ref}}$, typically refers to the initial model before reinforcement learning begins.

**Comment:** Empirically, we show that this variant exhibits surprisingly strong clinical reasoning capabilities even in models as small as 3B parameters. It is particularly effective when the model already possesses task-specific knowledge but struggles to apply it correctly in the EHR context. The trained model also demonstrates strong generalization across tasks. However, it cannot introduce new medical knowledge. As a result, when the model lacks essential domain understanding and seldom generates correct responses, RLVR training is prone to collapse.

### 2.2.2 REINFORCEMENT LEARNING WITH SFT WARM-UP

In this variant, we introduce a lightweight SFT warm-up phase before applying reinforcement fine-tuning, resulting in EHRMIND-SFT-RLVR.

In the SFT phase, we warm up the model using a small set of supervised examples $\{(i, x, o)\}$, where $o = (z, y)$ includes both a reasoning trace and the ground-truth answer. The model is trained to maximize the likelihood of generating the annotated output:

$$\mathcal{J}_{\text{SFT}}(\theta) = \mathbb{E}_{(i,x,o) \sim P(I,X,O)} \left[ \log \pi_\theta(o \mid i, x) \right]. \tag{3}$$

This initializes the model with a reasonable policy and structured outputs, which improves stability and accelerates RLVR. After SFT, we resume reinforcement training using the GRPO objective to further refine reasoning quality and outcome alignment.

**Comment:** Empirically, we show that this SFT warm-up can be performed with a relatively small number of examples, including those generated by an LLM, as RLVR can further refine the reasoning paths through trial and error. The warm-up phase effectively injects essential domain knowledge, providing RLVR with a stronger initialization. It also guides the model to generate reasoning traces with better clinical structure and interpretability, which are often lacking when using RLVR alone.

# 3  PILOT STUDY ON THE MEDCALC DATASET

As an initial exploration of RLVR for EHR-based reasoning tasks, we aim to investigate two key research questions Q1 and Q2 mentioned previously. We begin with a pilot study on the MEDCALC dataset (Khandekar et al., 2024), a benchmark designed to test medical calculation capabilities. MEDCALC offers a controlled yet realistic sandbox for probing LLM behaviors for several reasons:

- It tests several key clinical competencies, including medical knowledge, variable extraction, and clinical reasoning.
- Compared to multiple-choice medical exam questions (Hendrycks et al., 2021; Pal et al., 2022; Jin et al., 2020), it uses clinical notes as input, which better represents real-world decision-making scenarios.
- Unlike clinical predictive tasks such as risk prediction or diagnosis Harutyunyan et al. (2019); van de Water et al. (2024), it features explicit formulas and rule-based logic that enable interpretable evaluation.
- As a relatively new benchmark, it is less likely to overlap with pretraining corpora, allowing for a more reliable assessment of generalization capabilities.

## 3.1  EXPERIMENTAL SETUP

**Dataset.** In MEDCALC dataset (Khandekar et al., 2024), each instance includes a patient note and a calculator-specific instruction (e.g., "What is the patient's $CHA_2DS_2$-VASc score?"), with the objective of predicting a numeric, categorical, or datetime label. Calculators in MEDCALC are categorized into seven types: lab, physical, risk, diagnosis, severity, date, and dosage conversion. Specifically, each sample in the RLVR training set includes the instruction, patient context, and final answer, and is used for outcome-based optimization. More details can be found in the Appendix.

**Baselines.** We evaluate a diverse set of strong LLM baselines under the chain-of-thought (CoT) setting. These include open-source models such as LLaMA-3 (Grattafiori et al., 2024) (3B, 8B, 70B), as well as proprietary models including GPT-3.5 (Achiam et al., 2023) and GPT-4 (Ouyang et al., 2022b), Claude-3-Haiku (Anthropic, 2024b) and Claude-3.5-Sonnet (Anthropic, 2024a). We also include large-scale reasoning-tuned models such as o3-mini (OpenAI) and DeepSeek-R1 (Guo et al., 2025), which are explicitly designed to excel at multi-step reasoning.

**Training.** All our models are trained on top of the LLaMA-3-3B backbone. EHRMIND-RLVR is trained with RL only, using the RLVR training set and outcome-level reward feedback. EHRMIND-SFT is trained solely on the curated SFT set with reasoning supervision. EHRMIND-SFT-RLVR first applies SFT, and then continues training via RL on the RLVR training set. We hold out a small balanced validation set of 98 examples from the RLVR training data, with an equal number of instances from each calculator category. We report results on the test set using the checkpoint with the best validation performance. Hyperparameter configurations are provided in the Appendix.

**Evaluation Metric.** We report exact match accuracy as our evaluation metric. Each model is instructed to wrap its final answer within a special `<answer>...</answer>` tag. The span inside the tag is then extracted and compared against the ground-truth label. A prediction is considered correct only if the extracted string exactly matches the ground truth.

## 3.2  MAIN RESULTS

Table 1 shows the results of various models on the MEDCALC test set.

**Finding 1: A 3B model can exhibit strong clinical reasoning capabilities with RLVR alone.** When trained with our reinforcement-only recipe (EHRMIND-RLVR), a 3B LLaMA-3 model improves dramatically from its original zero-shot baseline of 9.74% to 41.26%. This absolute gain of over 30 points demonstrates that even without any supervised reasoning traces or domain-specific pretraining, pure outcome-driven reinforcement learning can substantially enhance task-specific performance. Remarkably, this 3B model outperforms powerful proprietary models like GPT-4 (37.92%) and Claude-3.5 Sonnet (41.18%), highlighting the effectiveness of our EHRMIND-RLVR recipe.

Table 1: Accuracy on the MEDCALC test set across seven calculator categories. **Bold** indicates the best result; underlined denotes the second and third best. EHRMIND *achieves state-of-the-art performance with only 3B parameters.* We provide further analysis for the Lab , Risk , and Sev. tasks in Finding 3 below, where EHRMIND (seemingly) ties/underperforms relative to baselines.

| Model | Size | Equation | | | | Rule-based | | | Overall |
|-------|------|----------|------|------|--------|------------|------|-------|---------|
| | | Lab | Phys. | Date | Dosage | Risk | Sev. | Diag. | |
| **Zero-shot (w/ reasoning)** | | | | | | | | | |
| LLaMA-3 (Grattafiori et al., 2024) | 3B | 8.87 | 15.83 | 1.67 | 7.50 | 7.92 | 8.75 | 8.33 | 9.74 |
| LLaMA-3 (Grattafiori et al., 2024) | 8B | 16.51 | 25.00 | 1.67 | 7.50 | 11.25 | 13.75 | 26.67 | 16.43 |
| LLaMA-3 (Grattafiori et al., 2024) | 70B | 33.94 | 66.25 | 25.00 | 20.00 | 18.33 | 16.25 | 36.67 | 35.53 |
| GPT-3.5 (Ouyang et al., 2022b) | - | 20.49 | 45.00 | 11.67 | 17.50 | 13.33 | 10.00 | 31.67 | 23.69 |
| GPT-4 (Achiam et al., 2023) | - | 26.30 | 71.25 | 48.33 | 40.00 | 27.50 | 15.00 | 28.33 | 37.92 |
| Claude-3-Haiku (Anthropic, 2024b) | - | 5.81 | 14.17 | 28.33 | 12.50 | 12.50 | 12.50 | 28.33 | 12.61 |
| Claude-3.5-Sonnet (Anthropic, 2024a) | - | 34.86 | 68.75 | 36.67 | 22.50 | 34.58 | 21.25 | 35.00 | 41.18 |
| o3-mini (OpenAI) | - | 48.01 | 71.25 | 28.33 | 37.50 | 36.25 | **30.00** | 25.00 | 46.42 |
| DeepSeek-R1 (Guo et al., 2025) | 671B | 53.21 | 73.75 | 8.33 | 42.50 | **38.75** | 10.00 | 50.00 | 48.13 |
| **Ours** | | | | | | | | | |
| EHRMIND-RLVR | 3B | 38.83 | **86.25** | 35.00 | 7.50 | 19.58 | 7.50 | 35.00 | 41.26 |
| EHRMIND-SFT (warm-up) | 3B | 30.88 | 49.19 | **55.00** | 75.00 | 26.25 | 16.25 | **63.33** | 37.82 |
| EHRMIND-SFT-RLVR | 3B | **55.66** | 81.25 | 53.33 | **80.00** | 22.08 | 18.75 | **63.33** | 51.96 |

**Finding 2: RL with SFT warm-up achieves state-of-the-art performance.** When preceded by a SFT warm-up on only ∼2k step-by-step reasoning examples, reinforcement learning yields even stronger results. Our 3B model (EHRMIND-SFT-RLVR) achieves an overall accuracy of 51.96%, outperforming all evaluated baselines—both open-source and proprietary. This includes models such as GPT-4 (37.92%), Claude-3.5 Sonnet (41.18%), and even large-scale reasoning models like o3-mini (46.42%) and DeepSeek-R1 (48.13%). These results highlight the effectiveness of outcome-oriented reinforcement learning: with only a lightweight SFT warm-up, a small LLM can surpass models that are orders of magnitude larger.

**Finding 3: RLVR struggles when essential domain knowledge is missing.** While our best model EHRMIND-SFT-RLVR achieves strong overall performance across categories, it still ties/underperforms models like o3-mini on *Lab*, *Risk*, and *Severity*. To understand this, we annotate each test instance as *seen* or *unseen* based on whether the required knowledge (e.g., clinical formula or concept) appears in the training set.

Among the seven categories, only *Lab*, *Risk*, and *Severity* test questions involve previously unseen clinical formulas or concepts. We further break down model performance across *seen* vs. *unseen* subsets. Compared to the initialized base model (LLaMA-3-3B), we observe that EHRMIND-SFT-RLVR yields strong gains on *seen* instances—often matching or exceeding o3-mini. However, on *unseen* instances, particularly in *Risk* and *Severity*, performance degrades and o3-mini maintains a clear advantage.

These results suggest that RLVR enhances reasoning by more effectively leveraging existing knowledge rather than introducing new information. This underscores the importance of constructing diverse and comprehensive training datasets for real-world clinical applications. Including data that spans a wide range of medical knowledge during training can significantly improve the robustness of outcome-oriented reinforcement learning in clinical reasoning tasks.

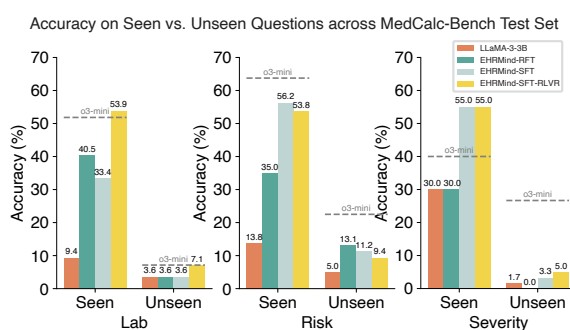

Figure 1: Comparison of accuracy on seen vs. unseen questions in the MEDCALC test set. Seen questions refer to those whose underlying medical knowledge was covered during training. EHRMIND-*SFT-RLVR yields strong gains on seen instances—often matching or exceeding o3-mini. However, it cannot inject medical knowledge not present in the training data, underscoring the importance of comprehensive coverage during training.*

## 3.3 When Is the SFT Warm-Up Necessary?

While reinforcement learning alone can significantly enhance performance, its effectiveness varies across task types. For example, in the *Dosage* category, a supervised warm-up proves critical for enabling meaningful improvement under RL. But when exactly is such a warm-up step necessary?

To investigate this, we analyze the relationship between a model's initial task competence and its performance gains from RLVR. Specifically, we use Pass@$k$ on the RLVR training set as a proxy for task solvability. We set $k = 12$ to match the number of rollout samples used during GRPO training. For each MEDCALC category, we compute Pass@12 for the LLaMA-3-3B base model and compare it to the accuracy improvement from LLaMA-3-3B to EHRMIND-RLVR on the held-out test set.

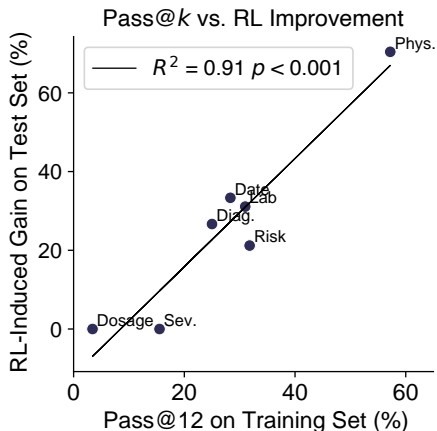

As shown in Figure 2, we observe a strong, statistically significant correlation ($R^2 = 0.91$, $p < 0.001$): categories with low Pass@12—such as *Dosage* (3.4%) and *Severity* (10.0%)—exhibit little to no improvement under pure RL. This suggests that **low initial Pass@12 predicts minimal RL improvement—highlighting when SFT warm-up is necessary for learning under sparse rewards.**

A concrete example is the *Dosage Conversion* category, which requires knowledge of domain-specific medication equivalencies (e.g., "What is the equivalent dose of Cortisone (PO) to PrednisoLONE (IV)?"). Such tasks involve specialized clinical knowledge that general-purpose LLMs like LLaMA-3-3B do not possess. Its Pass@12 on this task is just 3.42%, indicating that even with multiple attempts, the model rarely produces correct answers. Consequently, pure RL fails to yield meaningful improvement.

Figure 2: Relationship between Pass@$k$ on RLVR training set and RL-induced test set improvement. Each point corresponds to one category from MEDCALC. The x-axis shows the Pass@12 of LLaMA-3-3B on the RLVR training set, and the y-axis shows the resulting accuracy gain on the test set after applying EHRMIND-RLVR. *A strong correlation ($R^2 = 0.91$) suggests that low initial Pass@12 predicts minimal RL improvement—highlighting when SFT warm-up is necessary to unlock the full benefits of RLVR.*

On the other hand, introducing a lightweight supervised warm-up using ∼2k examples with step-by-step reasoning enables the model to acquire the missing task-specific knowledge. This warm-start allows reinforcement learning to take effect: EHRMIND-SFT-RLVR achieves the highest accuracy on *Dosage*, improving by more than 70 points over the base model. These findings underscore the practical utility of Pass@$k$ as a diagnostic tool: when it is low, an SFT warm-up is essential to unlock the full benefits of RLVR.

## 4 Evaluation on More Challenging EHR-Based Reasoning Tasks

### 4.1 Task Overview

To assess the generalizability of our findings beyond medical calculations, we further evaluate EHRMIND on additional EHR-grounded clinical tasks.

**Patient-Trial Matching.** This task is to determine whether a patient is eligible for a given clinical trial based on their medical record and the trial's eligibility criteria. The input consists of a synthetic patient note in free-text format and a textual description of the trial's inclusion and exclusion conditions. The model is prompted to predict one of three categories: *Excluded*, *Irrelevant*, or *Eligible*.

**Disease Diagnosis.** This task involves forecasting whether a specific disease (e.g., acute myocardial infarction) will occur for a patient within a defined time window. Given a temporally ordered sequence of structured clinical events—such as diagnoses, medications, and lab results—along with a prediction timestamp, the model must predict a binary outcome: 1 if the target disease will occur within the time window, and 0 otherwise.

Table 2: Performance on Patient-Trial Matching. EHRMIND-*SFT-RLVR achieves the best overall and per-class performance. SFT provides essential task-specific domain knowledge, while RLVR further refines the model's reasoning and decision boundaries.*

| Model | Size | Overall | | | Per-Class F1 Score | | |
|---|---|---|---|---|---|---|---|
| | | BACC | Macro F1 | Kappa | Excluded | Irrelevant | Eligible |
| **Zero-shot (w/ reasoning)** | | | | | | | |
| LLaMA-3 (Grattafiori et al., 2024) | 3B | 26.43 | 1.36 | 1.59 | 41.59 | 29.29 | 4.12 |
| LLaMA-3 (Grattafiori et al., 2024) | 8B | 32.90 | 2.36 | 7.04 | 40.72 | 20.79 | 42.35 |
| LLaMA-3 (Grattafiori et al., 2024) | 70B | 33.33 | 12.50 | 0.02 | 50.00 | 0.00 | 0.00 |
| GPT-4o (Achiam et al., 2023) | - | 38.16 | 24.77 | 7.24 | 50.35 | 25.64 | 23.09 |
| Claude-3-Haiku (Anthropic, 2024b) | - | 33.23 | 13.74 | 4.35 | 50.40 | 0.77 | 17.54 |
| Claude-3.5-Sonnet (Anthropic, 2024a) | - | 37.49 | 29.54 | 6.24 | 32.37 | 30.80 | 54.99 |
| o3-mini (OpenAI) | - | 39.60 | 5.78 | 10.18 | 24.34 | 39.36 | 57.76 |
| DeepSeek-R1 (Guo et al., 2025) | 671B | 35.02 | 23.68 | 3.39 | 10.17 | **43.95** | 40.58 |
| **Ours** | | | | | | | |
| EHRMIND-RLVR | 3B | 55.38 | 33.92 | 33.61 | 71.39 | 0.53 | 63.78 |
| EHRMIND-SFT (warm-up) | 3B | 41.71 | 35.15 | 20.95 | 44.29 | 37.55 | 58.78 |
| EHRMIND-SFT-RLVR | 3B | **63.14** | **44.47** | **44.70** | **71.44** | 34.61 | **71.82** |

## 4.2 EXPERIMENTAL SETUP

**Datasets.** For patient-trial matching, we use the TREC 2021 Clinical Trial dataset (Roberts et al., 2021), which contains synthetic patient records and eligibility criteria. For clinical event prediction, we use the diagnosis prediction tasks from the EHRSHOT benchmark (Wornow et al., 2023a), covering four diseases: Acute Myocardial Infarction, Hyperlipidemia, Hypertension, and Pancreatic Cancer. More details can be found in the Appendix.

**Training.** To avoid label imbalance and promote stable learning, we balance the class distribution within each task through downsampling. For each experiment, a held-out validation set is sampled from the training data for model selection. For SFT training, we construct intermediate reasoning traces for each task using GPT-generated outputs. More details can be found in the Appendix.

**Evaluation Metrics.** Since both tasks involve imbalanced classification problems, we report metrics commonly used in clinical ML to better reflect true performance. Following prior work (Grandini et al., 2020; Lin et al., 2023), we report: **Balanced Accuracy (BACC)**, **F1 Score** and **Cohen's Kappa**.

## 4.3 RESULTS ON PATIENT-TRIAL MATCHING

**Finding 1: Pass@12 highlights when SFT is needed to provide critical domain knowledge.** Following the findings from §3, we compute Pass@$k$ scores on the training set as a proxy to assess whether SFT may provide useful guidance for specific classes in the patient-trial matching task. However, with only a few discrete labels (e.g., three classes), classification tasks can be trivially guessed by chance, leading to inflated Pass@$k$ scores. To address this, we adopt a stricter metric—*Reliable Pass@12*—which discounts random guessing from the estimation by requiring the model to consistently produce correct predictions across multiple generations for the same input. This helps ensure that passed examples reflect meaningful model capabilities rather than by chance. Detailed computation is provided in the Appendix.

Using this approach, we randomly sample 100 training examples per class and find that LLaMA-3-3B performs reasonably well on the *Excluded* class (42%), but nearly fails on both *Irrelevant* (3%) and *Eligible* (0%). This suggests that SFT warm-up could provide valuable task-specific inductive bias—particularly for categories that require nuanced understanding and multi-step reasoning.

**Finding 2: SFT warm-up resolves class-specific weaknesses in RLVR.** From Table 2, we find that even with pure RLVR, EHRMIND-RLVR achieves strong gains across all overall metrics, outperforming all baselines. However, a closer look at the per-class performance reveals that EHRMIND-RLVR underperforms on the *Irrelevant* category, with an F1 score of just 0.53%. This trend can be traced back to the initialized LLaMA-3-3B model, whose Pass@12 on the training set is extremely low for the *Irrelevant* and *Eligible* classes (3% and 0%, respectively). This suggests that the model lacks the inductive bias or capacity to produce correct answers for these categories. In contrast, after applying supervised warm-up, EHRMIND-SFT-RLVR not only achieves the best overall performance across all metrics, but also consistently improves per-class performance.

Table 3: Performance on four disease diagnosis tasks. EHRMIND-*SFT-RLVR* achieves competitive or superior performance. Further analysis shows that it generates clinically meaningful rationales and generalizes more effectively across diagnostic conditions. [3]

| Model | Size | Acute MI | | | Hyperlipidemia | | | Hypertension | | | Pancreatic Cancer | | |
|---|---|---|---|---|---|---|---|---|---|---|---|---|---|
| | | BACC | F1 | Kappa | BACC | F1 | Kappa | BACC | F1 | Kappa | BACC | F1 | Kappa |
| **Zero-shot (w/ reasoning)** | | | | | | | | | | | | | |
| LLaMA-3 (Grattafiori et al., 2024) | 3B | 45.50 | 8.69 | 1.83 | 45.79 | 16.33 | -0.94 | 48.75 | 18.45 | 3.46 | 54.76 | 6.83 | 6.72 |
| LLaMA-3 (Grattafiori et al., 2024) | 8B | 58.53 | 15.78 | 4.43 | 53.27 | 22.36 | 4.15 | 56.95 | 24.97 | 6.59 | 67.29 | 10.44 | 6.35 |
| LLaMA-3 (Grattafiori et al., 2024) | 70B | 66.12 | 25.26 | 16.69 | 56.45 | 24.08 | 14.37 | 55.95 | 23.19 | 10.84 | 67.35 | 36.04 | 34.48 |
| GPT-4o (Achiam et al., 2023) | - | 63.28 | 20.73 | 10.93 | 56.11 | 24.47 | 9.15 | 62.21 | 30.15 | 14.84 | 70.67 | 37.50 | 35.66 |
| o3-mini (OpenAI) | - | 59.06 | 21.45 | 14.23 | 57.84 | 26.67 | 12.68 | 60.37 | 29.71 | 17.49 | 56.90 | 20.25 | 19.02 |
| DeepSeek-R1 (Guo et al., 2025) | 671B | 61.63 | 21.07 | 12.06 | 59.66 | 28.80 | 13.98 | 55.57 | 22.55 | 10.38 | 60.37 | 26.67 | 25.23 |
| **Ours** | | | | | | | | | | | | | |
| EHRMIND-RLVR | 3B | **68.17** | **28.04** | **19.99** | 59.91 | 29.32 | 15.34 | 60.43 | 30.92 | **21.21** | **81.79** | 31.08 | 28.27 |
| EHRMIND-SFT (warm-up) | 3B | 64.00 | 21.89 | 12.45 | 58.64 | 27.82 | 14.50 | 60.11 | 28.63 | 14.47 | 69.90 | 21.58 | 18.42 |
| EHRMIND-SFT-RLVR | 3B | 67.44 | 26.98 | 18.73 | **62.50** | **31.24** | **15.42** | **65.45** | **33.85** | 19.76 | 80.41 | 32.17 | 29.49 |

## 4.4 RESULTS ON DISEASE DIAGNOSIS

**Finding 1: EHRMIND achieves improved accuracy across tasks.** As shown in Table 3, both EHRMIND-RLVR and EHRMIND-SFT-RLVR achieve strong overall results across four clinical prediction tasks. Notably, EHRMIND-SFT-RLVR outperforms EHRMIND-RLVR on three of the four tasks and achieves the best performance on Hyperlipidemia and Hypertension. This highlights the value of even limited and noisy reasoning supervision during pretraining, which provides a useful initialization and stabilizes downstream RL optimization.

One exception is the Acute MI task, where EHRMIND-SFT-RLVR slightly underperforms EHRMIND-RLVR (67.44% vs. 68.17% BACC). We attribute this to two potential factors: (1) the SFT dataset for Acute MI is relatively small—only several hundred examples—due to the limited number of positive (diagnosed) cases available in the training set. This data scarcity may have constrained the model's ability to learn effective reasoning patterns specific to this condition; and (2) the binary classification structure may allow RL to hack rewards by exploiting shallow decision rules, which may suffice for performance but do not necessarily encourage robust clinical reasoning.

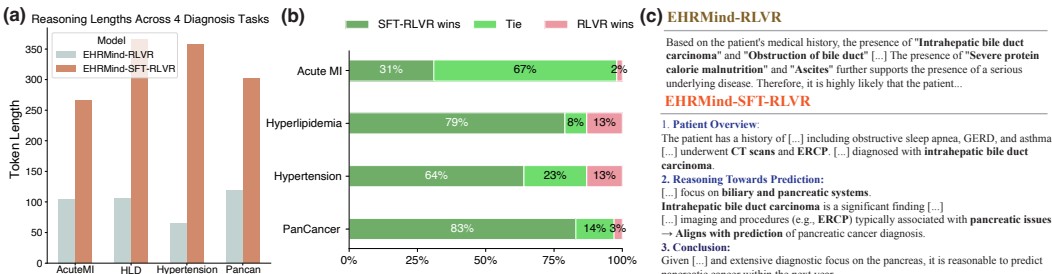

Figure 3: Quantitative and qualitative analysis of reasoning improvements from SFT warm-up on disease diagnosis tasks. **(a)** Reasoning lengths averaged over 100 sampled test examples per condition. **(b)** Pairwise evaluation results of comparing reasoning quality between EHRMIND-SFT-RLVR and EHRMIND-RLVR. **(c)** Case study from the Pancreatic Cancer task. EHRMIND-*SFT-RLVR* demonstrates more structured and clinically aligned reasoning, explicitly connecting diagnostic evidence to the prediction, whereas EHRMIND-*RLVR* lacks such specificity.

**Finding 2: SFT warm-up prevents reasoning collapse during RLVR.** To further understand the impact of the SFT warm-up, we analyze the length of reasoning trace generated by each model. As shown in Figure 3(a), EHRMIND-RLVR consistently produces much shorter rationales across all tasks. This aligns with recent findings in the literature (Li et al., 2025; He et al., 2025), where RL-based training for classification tasks often leads to a collapse in the reasoning process: models simply omit reasoning steps and directly output final predictions. Such behavior is inadequate for clinical decision-making, where interpretability and transparency are critical. In contrast, SFT warm-up in EHRMIND-SFT-RLVR preserves detailed reasoning structures throughout RL training.

---

[3] Per the EHRSHOT (Wornow et al., 2023a) License, we used GPT-4o and o3-mini via Microsoft Azure's HIPAA-compliant platform under a signed BAA. We did not use Claude, as it does not offer a HIPAA-compliant deployment.

**Finding 3: SFT-RLVR produces more coherent and clinically aligned rationales.** We further assess reasoning quality through a pairwise evaluation using GPT-4o. For each task, we randomly sample 100 test examples and leverage GPT-4o to compare the reasoning text generated by EHRMIND-SFT-RLVR and EHRMIND-RLVR. Figure 3(b) shows that across all four tasks, GPT-4o consistently prefers the reasoning from EHRMIND-SFT-RLVR, indicating more coherent and clinically meaningful explanations.

A case study from the Pancreatic Cancer task (Figure 3(c)) further illustrates these differences. EHRMIND-RLVR simply produces a generic explanation, while EHRMIND-SFT-RLVR delivers a structured rationale that connects diagnostic evidence (e.g., ERCP, CT scan) to the prediction, and references relevant anatomical systems. This example highlights how SFT warm-up contributes to clinically appropriate, transparent reasoning, which is crucial for real-world adoption.

**Finding 4: RL-based optimization generalizes better across diagnostic tasks.** We assess the generalizability of different training paradigms by evaluating cross-task performance. Specifically, we train models on the Hyperlipidemia task and test how well they generalize to the other three diagnosis targets. As shown in Figure 4, SFT-only training on Hyperlipidemia (SFT-Hyperlip) yields weak transfer. In contrast, both RLVR and SFT-RLVR demonstrate substantially better generalization, even outperforming in-distribution SFT models in some cases. These findings suggest that RL-based optimization encourages the development of clinical reasoning patterns that transfer across diagnostic contexts.

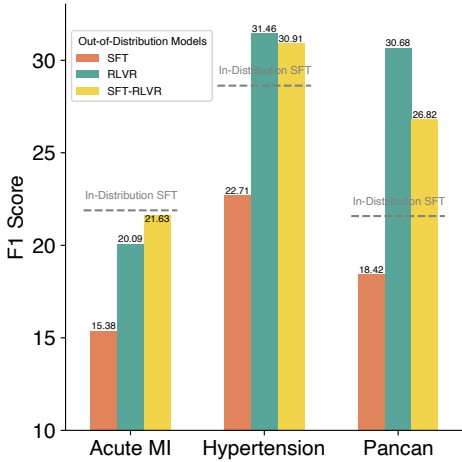

Figure 4: Comparison of cross-task generalization performance. Each model is trained exclusively on Hyperlipidemia data and evaluated on out-of-distribution diagnosis tasks. *RLVR generalizes better than SFT models, suggesting reinforcement learning promotes transferable reasoning patterns across clinical conditions.*

**Finding 5: Pass@12 remains a reliable indicator of SFT necessity.** We compute Pass@12 on the training set for each clinical event prediction task by sampling 100 examples per task. Table 4 breaks down performance by label class and reveals a consistent trend: across all four tasks, the model performs substantially better on the negative class, while consistently struggling to generate the correct positive label. This asymmetry reflects the LLaMA-3-3B model's limited ability to discover the correct decision boundary for rare or nuanced clinical outcomes. Particularly, tasks with low overall Pass@12 scores—such as Hyperlipidemia and Hypertension—show greater performance gains from SFT warm-up in Table 3.

Table 4: Training Set Pass@12 on Clinical Event Prediction Tasks. (0 = No Diagnosis, 1 = Diagnosis). *Tasks with low overall Pass@12 scores—such as Hyperlipidemia and Hypertension—show greater performance gains from SFT warm-up in Table 3.*

| Task | Class 0 | Class 1 | Overall |
|---|---|---|---|
| Acute MI | 0.47 | 0.06 | 0.265 |
| Hyperlipidemia | 0.18 | 0.04 | 0.110 |
| Hypertension | 0.31 | 0.09 | 0.200 |
| Pancreatic Cancer | 0.79 | 0.13 | 0.460 |

## 5 CONCLUSION

In this work, we propose EHRMIND, a practical recipe for adapting LLMs to EHR-based reasoning tasks via RLVR. Even without supervised reasoning traces, RLVR alone enables a 3B model to outperform significantly larger proprietary LLMs. However, we show that a lightweight SFT warm-up is critical when the model lacks initial task competence, especially in tasks requiring specialized domain knowledge. Across diverse benchmarks—including medical calculations, patient-trial matching, and disease diagnosis—EHRMIND consistently achieves state-of-the-art performance, generates more coherent and clinically grounded rationales, and exhibits stronger cross-task generalization. We further validate the utility of Pass@k as a diagnostic tool to identify when SFT warm-up is necessary, highlighting its value for guiding efficient training strategies in real-world healthcare settings.

ETHICS STATEMENT

In this study, we evaluate our methods on three datasets: MedCalc-Bench (Khandekar et al., 2024), TREC Clinical Trials (Roberts et al., 2021), and EHRSHOT (Wornow et al., 2023a). Both MedCalc and TREC Clinical Trials contain entirely synthetic patient records and do not involve real patient data, eliminating concerns regarding privacy or personal health information. For the EHRSHOT dataset, which is derived from real EHRs, we followed the licensing and usage restrictions outlined in the EHRSHOT License (Wornow et al., 2023a). Specifically, all experiments involving closed-source models on the EHRSHOT dataset were conducted through Microsoft Azure's HIPAA-compliant cloud infrastructure under a signed Business Associate Agreement (BAA), in accordance with the EHRSHOT License (Wornow et al., 2023a).

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

**Contents of Appendix**

## A    DECLARATION OF LLM USAGE

We use large language models (LLMs) in the following ways throughout our study:

1. We employ LLMs to assist in constructing supervised warm-up datasets for both the Patient-Trial Matching and EHRSHOT tasks. Details are provided in Section F.

2. We include LLMs as baseline models for evaluation on all three benchmarks: MedCalc-Bench, TREC Clinical Trials, and EHRSHOT.

3. We use LLMs as evaluators to compare the quality of model-generated reasoning chains in the EHRSHOT Clinical Event Prediction task. See Section F.3.4.

All usage was conducted responsibly and within the scope of model licenses and data-sharing agreements.

## B    BROADER IMPACTS AND LIMITATIONS

**Broader Impacts**    Our work explores how reinforcement learning with verifiable rewards (RLVR) can enable more accurate and interpretable clinical reasoning in large language models. By focusing on EHR-grounded tasks, EHRMIND has the potential to support real-world medical applications such as diagnosis prediction and trial matching. At the same time, we emphasize that RLVR is not universally effective—especially when models lack prior domain knowledge—highlighting the need for careful design, supervision, and evaluation in clinical deployments.

**Limitations**    While EHRMIND demonstrates strong performance across multiple EHR-grounded tasks, several limitations remain.

First, when patient histories contain long sequences of events, standard LLMs face challenges due to extremely long context length. Addressing this may require hybrid or multimodal approaches that encode individual clinical events as structured representations rather than feeding raw text directly into the model, as explored in prior work such as Wu et al. (2024).

Second, although RLVR bypasses the need for annotated reasoning traces by supervising solely on outcomes, it still heavily depends on the scale and diversity of training data. We observe that larger and more diverse datasets significantly improve the stability and effectiveness of RL optimization. In contrast, limited or narrow training distributions may restrict the model's ability to generalize, especially in clinically diverse or rare scenarios.

Finally, our approach relies on rule-based reward functions, which may not be readily available or easily constructed for many real-world clinical tasks. Unlike classification or medical calculation—where correctness can be directly determined by comparing model outputs to ground-truth labels or reference values—tasks such as medical report generation or summarization require evaluation across multiple dimensions (e.g., factual accuracy, coherence, clinical appropriateness). Designing effective reward functions for such tasks is substantially more challenging, often requiring expert-defined criteria or multi-faceted scoring mechanisms.

## C    SAFEGUARDS AND LICENSES FOR ASSETS

**Safeguards**    We train models on three datasets: MedCalc, TREC Clinical Trials, and EHRSHOT. For MedCalc and TREC, both of which use synthetic or publicly available clinical data, we believe that releasing models trained on them poses minimal risk of misuse. In contrast, since EHRSHOT contains real but de-identified patient data, we do not release any models trained on it to avoid potential misuse or unintended privacy concerns.

**Licenses for Assets**    Our paper uses three existing datasets: MedCalc-Bench, TREC Clinical Trials, and EHRSHOT. All assets are used in accordance with their respective licenses and terms of use. MedCalc-Bench is released under the CC-BY-SA 4.0 license and is publicly available at `https://huggingface.co/datasets/ncbi/MedCalc-Bench-v1.0`. TREC Clinical Trial dataset is in the public domain with no copyright restrictions, which can be found at `https:`

//www.trec-cds.org/2021.html. EHRSHOT is licensed under the EHRSHOT Data Set License 1.0, modeled after PhysioNet Version 1.5.0. Its use is restricted to lawful scientific research under privacy and security requirements. We accessed the dataset via `https://redivis.com/datasets/53gc-8rhx41kgt`. We followed all licensing terms and data usage restrictions as specified by the dataset providers.

## D  RELATED WORK

**Reinforcement Learning for Language Models.**   Reinforcement learning (RL) has become a foundational approach for aligning LLMs with human preferences and task-specific objectives. Early efforts, such as Reinforcement Learning from Human Feedback (RLHF), focused on modeling human preferences for dialogue systems (Ouyang et al., 2022a; Bai et al., 2022). More recent approaches—including Direct Preference Optimization (DPO) (Rafailov et al., 2024) and Reinforcement Learning from AI Feedback (RLAIF) (Lee et al., 2024)—investigate more scalable and efficient supervision signals. A parallel line of work, Reinforcement Learning with Verifiable Rewards (RLVR), replaces human preference modeling with rule-based reward functions to promote correctness in structured domains such as mathematics and programming (Shao et al., 2024). Our work extends RLVR to the clinical domain, focusing on EHR-based reasoning tasks where reward signals can be programmatically derived from clinical formulas, eligibility criteria, and diagnostic consistency. To our knowledge, this represents one of the first applications of RLVR in EHR-based reasoning tasks, differing from prior RL-based efforts that primarily target medical exam-style QA (Zhang et al., 2025) or multimodal clinical VQA (Lai et al., 2025).

**LLMs for Clinical Reasoning.**   LLMs are increasingly being explored for a range of clinical tasks, including summarization, question answering, and diagnosis (Singhal et al., 2023; Nori et al., 2023). Many of these systems rely on prompt engineering or supervised fine-tuning using task-specific datasets (Wu et al., 2024; Cui et al., 2025; Fleming et al., 2023). Instruction tuning, in particular, has emerged as an effective paradigm for aligning models with diverse downstream tasks. In the clinical domain, instruction datasets like MEDALIGN (Fleming et al., 2023) and MIMIC-INSTR (Wu et al., 2024) enable broader task coverage. However, these approaches often require curated or synthetic annotations that may not transfer well across settings. Our method complements this line of research by exploring whether outcome-driven feedback through RLVR can induce clinical reasoning capabilities without dense intermediate supervision. To support scenarios where domain-specific annotations are limited, we employ a lightweight SFT warm-up phase, using a small number of annotated examples to bootstrap RLVR training.

## E  RELIABLE PASS@K

Standard Pass@$k$ metrics are often used to approximate a model's capability to solve a task through sampling. In our work, we adopt Pass@$k$ on the training set as a proxy for estimating whether the model possesses sufficient prior competence to benefit from RLVR alone. However, for tasks with small discrete label spaces—such as classification—standard Pass@$k$ can be unreliable: the model may guess the correct answer by chance across multiple rollouts, leading to an overestimation of true task proficiency.

To address this issue, we introduce **Reliable Pass@$k$**, a stricter alternative designed to more accurately reflect model competence on classification or discrete decision problems. Our method is motivated by the observation that confident, consistent predictions are more indicative of real model understanding than occasional lucky guesses.

**Definition.**   Given $k$ generated outputs per example, let $c$ be the number of times the correct prediction appears. We define:

- A sample is considered **passed** if $c \geq \tau_p$, where $\tau_p = \lceil k/C \rceil + 2$ and $C$ is the number of possible classes. This threshold ensures that the correct answer appears with significantly higher frequency than would be expected from random guessing alone.

Table 5: Number of examples per category across different data splits. Categories are grouped into equation-based and rule-based calculators. Abbreviations: Phys. = Physical; Sev. = Severity; Diag. = Diagnosis; Dosoage = Dosage Conversion.

| Data Split | Equation-based | | | | Rule-based | | | Total |
|---|---|---|---|---|---|---|---|---|
| | Lab | Phys. | Date | Dosage | Risk | Sev. | Diag. | |
| RLVR Training Set | 3,124 | 4,836 | 240 | 160 | 1,229 | 77 | 387 | 10,053 |
| SFT Training Set | 287 | 287 | 240 | 160 | 672 | 77 | 287 | 2,010 |
| Test Set | 327 | 240 | 60 | 40 | 240 | 60 | 80 | 1,047 |

- Let $\mathcal{D}$ be the empirical distribution of predicted labels across $k$ outputs. We compute its entropy as:

$$H(\mathcal{D}) = -\sum_{x \in \mathcal{D}} p(x) \log p(x) \tag{4}$$

  where $p(x)$ denotes the relative frequency of label $x$ among the $k$ outputs.

- A **confident pass** is recorded only if the sample passes ($c \geq \tau_p$) and the entropy is low: $H(\mathcal{D}) < \tau_e$, where $\tau_e = 0.8 \log C$ is an entropy threshold scaled by the class count. This ensures that correct predictions are not only frequent but also made with high confidence (i.e., low label uncertainty).

**Usage in Experiments.** In our experiments, we apply Reliable Pass@$k$ to tasks with discrete label spaces, where standard Pass@$k$ may overstate the model's true ability due to random guessing. Specifically, we use Reliable Pass@$k$ for the *Patient-Trial Matching* and *Diagnosis Prediction* tasks, both of which are multi-class classification problems. Within the MEDCALC benchmark, we apply Reliable Pass@$k$ to three rule-based categories: *Diagnosis*, *Risk*, and *Severity*, as their label spaces are discrete.

For these three tasks in MEDCALC, to instantiate the entropy threshold $\tau_e$, we estimate the number of distinct labels $C$ for each task by enumerating the unique ground-truth values in the training data. Based on this heuristic, we set $C = 7$ for *Diagnosis* and *Severity*, and $C = 21$ for *Risk*. These values are then used to compute the entropy threshold $\tau_e = 0.8 \log C$ in our Reliable Pass@$k$ computation.

**Runtime Efficiency for Pass@$k$ and Reliable Pass@$k$.** Both Pass@$k$ and Reliable Pass@$k$ require sampling $k$ outputs per training example. To understand the practical feasibility of using these metrics for deciding whether SFT warm-up before RLVR is necessary, we analyze their runtime efficiency on patient-trial matching. Specifically, computing Pass@12 across 300 training samples (100 per class) takes approximately **1 hours**. In contrast, RLVR training on patient-trial matching takes over **7 hours** to reach peak validation performance. Full-scale RLVR training on the entire dataset typically requires nearly **21 hours** for one epoch.

This analysis highlights the value of Pass@$k$ as a lightweight alternative to full-scale RL. When Pass@$k$ or Reliable Pass@$k$ scores are low, this serves as an early indication that the base model lacks sufficient task-specific competence—prompting the use of SFT warm-up.

# F ADDITIONAL EXPERIMENT AND RESULT DETAILS

## F.1 MEDCALC

### F.1.1 DATASET DETAILS

We evaluate our method on the MEDCALC-BENCH dataset (Khandekar et al., 2024), which spans a diverse set of clinical calculators. Table 5 summarizes the number of examples per category across different data splits. The RLVR training set corresponds to the full training set provided by MedCalc. The SFT training set is a randomly selected subset, with class balancing to ensure representation across all categories. For each SFT sample, we use the official step-by-step explanation as the reasoning process for supervision.

The prompt format used for both SFT and RLVR is provided in Table 6. Each instance contains a clinical note and a calculator-specific question, and the model is instructed to reason in <think> tags and output a final answer in structured <answer> JSON format.

Table 6: Prompt template used for MedCalc tasks.

| Prompt Template (SFT and RLVR) |
|---|

```
<|begin_of_text|><|start_header_id|>system<|end_header_id|>
You are a helpful assistant.  You first think about the
reasoning process in the mind and then provide the user with
the answer.
<|eot_id|>
<|start_header_id|>user<|end_header_id|>
You are a helpful assistant for calculating a score for a given
patient note.  Please think step-by-step to solve the question
and then generate the required score.
Here is the patient note:
{note}
Here is the task:
{question}
Please show your entire reasoning process in **a single**
<think> </think> block (do not open or close the tag more than
once).
Your final response must be in JSON format within <answer>
</answer> tags.  For example,
<think>
[entire reasoning process here]
</think>
<answer>
{
"answer":  str(short_and_direct_answer_of_the_question)
}
</answer>
Do not output anything after the </answer> tag.
<|eot_id|>
<|start_header_id|>assistant<|end_header_id|>
Let me solve this step by step.
<think>
```

### F.1.2 SEEN VS. UNSEEN QUESTION TYPES

To evaluate generalization, we classify each test example as either *seen* or *unseen*, depending on whether its underlying medical knowledge was encountered during training. Table 7 shows the number and proportion of unseen questions per category.

For the *Dosage* category, we do not rely on exact string matching to determine whether a question is seen or unseen. Unlike other categories, dosage-related questions follow a generalizable template—e.g., "Given a dose of Drug A, what is the equivalent dose of Drug B?"—which requires applying a drug-specific conversion factor. Importantly, all 8 drugs involved in the test set appear in the training set. This means that, although specific drug pairs in test questions may not exactly match those seen during training, the model should have been exposed to the relevant conversion factors for all drugs. As such, we treat all dosage questions as *seen* from a knowledge coverage perspective.

Table 7: Number of unseen and total test examples for each category in the MedCalc benchmark. A question is considered *unseen* if its underlying medical knowledge was not encountered during training.

| Category | Lab | Physical | Date | Dosage | Risk | Severity | Diagnosis |
|---|---|---|---|---|---|---|---|
| Unseen | 4 | 0 | 0 | 0 | 8 | 3 | 0 |
| Total | 19 | 12 | 3 | 16 | 12 | 4 | 3 |

### F.1.3 IMPLEMENTATION DETAILS

We implement our training pipeline for MEDCALC using the open-source VeRL (Sheng et al., 2024) framework, and conduct all experiments on two NVIDIA A100 GPU with 80GB memory. We apply Group Relative Policy Optimization (GRPO) as our reinforcement learning algorithm. The base

language model is initialized from `LLaMA-3-3B-Instruct` and optimized with KL-regularized policy gradients to prevent policy collapse. We use a low-variance KL penalty with coefficient $\lambda_{\text{KL}} = 0.001$.

Each input prompt is rolled out $k = 12$ times using top-$p$ sampling ($p = 0.95$) and temperature 0.6. Rollouts are performed using vLLM with GPU memory utilization capped at 40% to ensure stability. To support large-batch training, we enable gradient checkpointing and adopt Fully Sharded Data Parallelism (FSDP) with parameter, gradient, and optimizer offloading. The global mini-batch size is 128, with micro-batch size 4. All training runs span 2 epochs with a learning rate of $1 \times 10^{-6}$. We truncate each patient note and question to a maximum combined prompt length of 2048 tokens, with generated outputs truncated at 1500 tokens.

### F.2 TREC CLINICAL TRIAL

#### F.2.1 DATASET DETAILS

We use the dataset released in the TREC 2021 Clinical Trials Track (Roberts et al., 2021), which contains labeled instances of patient-trial pairs categorized into three classes: *Eligible*, *Excluded*, and *Irrelevant*. We randomly partition the data and ensure approximate class balance within each set. We finally obtain a split of 13,011 training examples and 10,068 test examples.

For model training, we further (1) set aside a validation split from the training data, and (2) filter out training examples whose combined input length exceeds 1024 words. This yields a filtered dataset comprising 11,258 training examples, 1,000 validation examples, and 10,068 test examples.

#### F.2.2 SFT DATA CONSTRUCTION

We first analyze model behavior using Reliable Pass@$k$ as introduced in Section E. We observe that model performance is notably poor on certain classes, indicating the base model lacks sufficient task-specific competence. In such cases, as discussed in MedCalc part, SFT warm-up can help inject inductive bias and bootstrap subsequent RLVR training.

To obtain SFT data, we follow the reasoning data generation paradigm proposed by Jiang et al. (2024). Specifically, we prompt GPT-4o to generate detailed step-by-step reasoning traces for a given patient-trial pair and its ground-truth label. The LLM is asked to explain its reasoning without revealing the label in the rationale itself.

We sample 3,000 patient-trial pairs from the training set (balanced across three classes) and generate reasoning chains for each using GPT-4o. Only examples with high-confidence outputs (e.g., excluding generations marked "Not Confident") are retained. These generations are then converted into input-output pairs suitable for SFT training, following our prompt schema in 8. The resulting 2,998 SFT data are used to warm-start our EHRMIND pipeline before RLVR optimization.

#### F.2.3 IMPLEMENTATION DETAILS

The reinforcement learning setup for the TREC CLINICAL TRIAL task closely follows the same configuration used for MEDCALC, as described in Appendix F.1.3. We adopt the same base model, optimizer settings, sampling configuration, and PPO parameters via the `VeRL` codebase. The only exception lies in the input/output length constraints: for patient-trial matching, we set `max_prompt_length = 1500` and `max_response_length = 2048`.

### F.3 EHRSHOT

#### F.3.1 DATASET DETAILS

EHRSHOT is an EHR dataset sourced from the Stanford Medicine Research Data Repository (Datta et al., 2020), which includes electronic health records from both Stanford Health Care (primarily adult care) and Lucile Packard Children's Hospital (primarily pediatric care). The publicly released version comprises 6,739 patients and approximately 41 million events. These events include demographics (e.g., age, sex, race), diagnoses, procedures, laboratory results, medication prescriptions, and other coded clinical observations. All events are temporally ordered.

Table 8: Prompt template for generating SFT data for patient-trial matching. The LLM receives a structured query containing the task description, patient note, clinical trial information, and the ground-truth label. It is then asked to generate a step-by-step reasoning chain.

---

**Prompt Template (SFT Data for Patient-Trial Matching)**

---

```
Given the following task description, patient EHR context,
clinical trial information, and the ground truth eligibility
label, provide a step-by-step reasoning process that leads to
the correct prediction:
=========================================
# Task
Patient-Trial Matching Task:
Objective:  Determine whether a patient is eligible for a given
clinical trial based on the patient's medical note and the
trial's inclusion/exclusion criteria.
Labels:
0) Irrelevant (patient does not have sufficient information to
qualify for the trial);
1) Excluded (patient meets inclusion criteria, but is excluded
on the grounds of the trial's exclusion criteria); and
2) Eligible (patient meets inclusion criteria and exclusion
criteria do not apply).
Key Considerations:
- Carefully **evaluate each inclusion and exclusion criterion
individually**.
- For each criterion, determine whether the patient **clearly
satisfies**, **clearly violates**, or has **insufficient
information**.
=========================================
# Patient EHR Note
{patient_context}
=========================================
# Clinical Trial
{trial_info}
=========================================
# Ground Truth
{ground_truth}
=========================================
Please provide a step-by-step reasoning process that leads to
the correct prediction based on the patient's EHR context.
**The reasoning chain should follow this structured format:**
1.  **Patient and Clincial Trial Overview**:  Go over the key
information in the patient's EHR context and the clinical
trial criteria, with the **Key Considerations** from the task
description in mind.
2.  **Reasoning Towards Prediction**:  Integrate the above
information to logically reason towards the predicted outcome.
3.  **Conclusion**:  Summarize the reasoning and state the
prediction without mentioning the ground truth label.
The reasoning should be comprehensive, medically sound, and
clearly explain how the patient's information leads to the
predicted outcome.
**Important Notes:**
- **Do not mention the ground truth label in the reasoning
process**.
- Use the relevant knowledge as needed, but **the main focus
should be on the patient's EHR context and the clinical trial**.
After generating the reasoning chain, please review it and
indicate your confidence in the reasoning chain at the end.
Options of confidence:  [Very Confident, Confident, Neutral, Not
Confident, Very Not Confident.]
**Output Format:**
# Reasoning Chain #
1.  **Patient and Clincial Trial Overview**:
[YOUR OUTPUT]
2.  **Reasoning Towards Prediction**:
[YOUR OUTPUT]
3.  **Conclusion**:
[YOUR OUTPUT]
# Confidence #
[CONFIDENCE (choose one:  "Very Confident", "Confident",
"Neutral", "Not Confident", "Very Not Confident")]
```

---

EHRSHOT defines 15 tasks, broadly categorized into the following four groups: (1) Operational Outcomes, (2) Anticipating Lab Test Values, (3) Assignment of New Diagnoses, and (4) Anticipating Chest X-ray Findings. Due to computational constraints, we focus on four binary classification tasks from the Assignment of New Diagnoses category: Acute Myocardial Infarction (MI), Hyperlipidemia, Hypertension, and Pancreatic Cancer. These tasks involve predicting the first diagnosis of a disease. The prediction time is set to 11:59 p.m. on the day of discharge from an inpatient visit, and any diagnosis that occurs within 365 days post-discharge is considered a positive outcome.

### F.3.2 EHR SERIALIZATION FOR LLM INPUT

To convert structured EHR event streams into a format suitable for large language models (LLMs), we transform each patient's event sequence into a textual representation. Events are grouped by timestamp, and all events occurring at the same timestamp are listed together. The format is illustrated in Table 9. This serialized representation is tokenized and provided as input to the LLM. However,

Table 9: Textual serialization format of EHR event sequences used as input to the LLM.

| **EHR Event Text Format** |
| --- |
| ```
timestamp
    event 1 name
    event 2 name
    event 3 name
``` |
| **Example:** |
| ```
2020-03-15 19:55:00
    Chest pain
    Metoprolol
    Electrocardiogram ordered
2020-03-16 08:20:00
    Echocardiogram performed
    Beta blocker therapy continued
``` |

a complete EHR history can contain over 10,000 events for a single patient, which far exceeds the context window limitations of most LLMs. Figure 5 shows the average number of events per patient by event type in the EHRSHOT dataset. Notably, `measurement` events dominate in volume, followed by `observation`, `drug_exposure`, and `note` events.

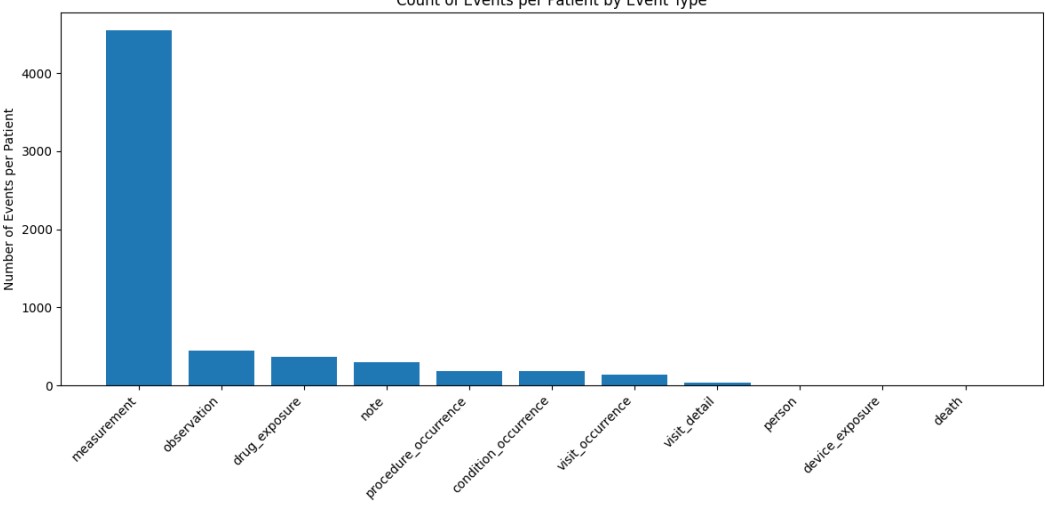

Figure 5: Average number of events per patient by event type in the EHRSHOT dataset. Measurement events constitute the largest portion of EHRs, followed by observations, drug exposures, and clinical notes.

To reduce sequence length while preserving clinical relevance, we perform an ablation study to evaluate the predictive utility of individual event types. Each patient is represented as a high-dimensional sparse vector using a bag-of-events approach, where each dimension corresponds to a unique event code (e.g., ICD, CPT, LOINC, RxNorm). We then train XGBoost classifiers for each of the four diagnosis prediction tasks using vectors composed solely of one event type at a time. Figure 6 presents the AUROC scores from models trained on each event type in isolation. The results highlight the differential importance of event types: `procedure_occurrence`, `condition_occurrence`, and `measurement` consistently yield strong performance. For instance, `procedure_occurrence` alone achieves an AUROC of 0.82 for pancreatic cancer prediction, comparable to full-feature models in some cases. Despite their high utility, `measurement` events are far more frequent than other types

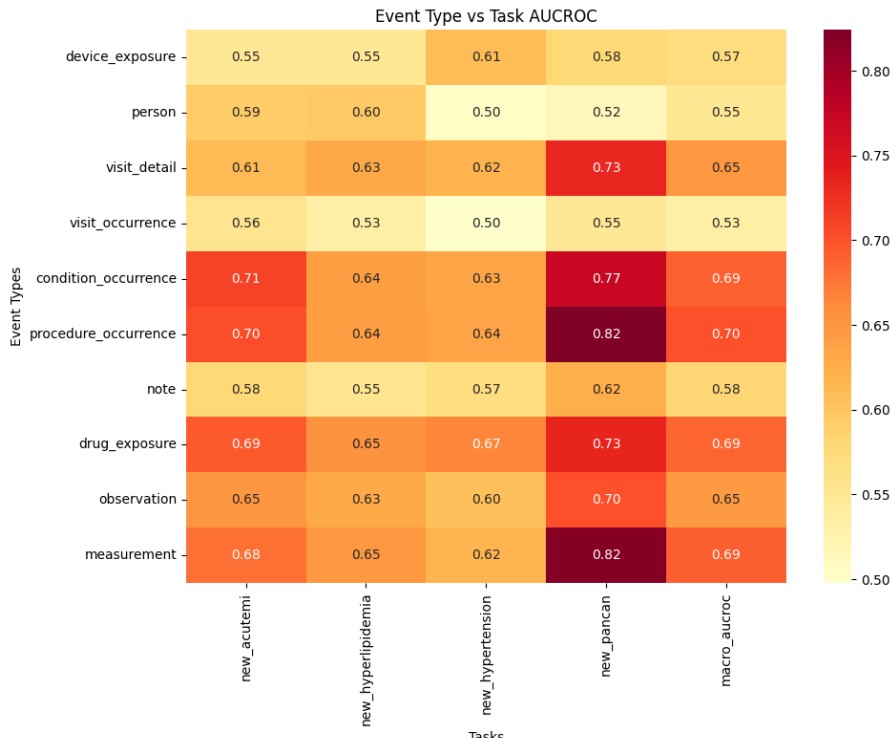

Figure 6: AUROC of XGBoost classifiers trained on individual event types across the four diagnosis prediction tasks in EHRSHOT. `Procedure_occurrence`, `condition_occurrence`, and `measurement` demonstrate the highest predictive value.

(Figure 5), posing challenges for models with constrained input length. To balance predictive signal and sequence length, we prioritize event types with high information density and moderate frequency. Based on this trade-off, we select `condition_occurrence` and `procedure_occurrence` as the core event types for constructing compact, LLM-compatible patient representations.

### F.3.3 SFT DATA CONSTRUCTION

For the EHRSHOT benchmark, we follow a similar strategy as in patient-trial matching to construct SFT warm-up data. We synthesize SFT training data using GPT-4o by prompting it with the full patient history, a prediction timestamp, and a label to generate medically grounded reasoning paths. We discard low-confidence completions and only retain examples rated by GPT-4o as "Confident" or "Very Confident."

For each of the four diagnosis prediction tasks—`Acute MI`, `Hyperlipidemia`, `Hypertension`, and `Pancreatic Cancer`—we sample examples from the training set while ensuring a balanced number of positive and negative cases. The final number of SFT examples per task is: 350 for Acute MI, 402 for Hyperlipidemia, 354 for Hypertension, and 304 for Pancreatic Cancer.

An example prompt used for the `Acute MI` task is shown in Table 10. All examples follow the same structure across tasks, with only the task description updated accordingly.

Table 10: Prompt template for generating SFT data for EHR-based diagnosis prediction (e.g., Acute MI, Pancreatic Cancer). The LLM is provided with task description, prediction timestamp, patient context, and ground-truth label, and is instructed to generate a reasoning chain.

---

**Prompt Template (SFT Data for EHR-Based Diagnosis Prediction)**

---

```
Given the following task description, patient EHR context,
prediction timestamp and ground truth label, provide a
step-by-step reasoning process that leads to the correct
prediction:
========================================
# Task
Acute Myocardial Infarction Prediction Task:
Objective:  Predict whether the patient will have her first
diagnosis of an acute myocardial infarction within the next
year.
Labels:  1 = first diagnosis within 1 year, 0 = no diagnosis
within 1 year
========================================
# Prediction Timestamp
{prediction_timestamp}
========================================
# Patient EHR Context
{patient_context}
========================================
# Ground Truth
{ground_truth}
========================================
Please provide a step-by-step reasoning process that leads to
the correct prediction based on the patient's EHR context and
prediction timestamp.
**The reasoning chain should follow this structured format:**
1.  **Patient Overview**:  Go over the key information in the
patient's EHR context.
2.  **Reasoning Towards Prediction**:  Integrate the above
information to logically reason towards the predicted outcome.
3.  **Conclusion**:  Summarize the reasoning and state the
prediction without mentioning the ground truth label.
The reasoning should be comprehensive, medically sound, and
clearly explain how the patient's information leads to the
predicted outcome.
**Important Notes:**
- **Do not mention the ground truth label in the reasoning
process**.
- Use the relevant knowledge as needed, but **the main focus
should be on the patient's EHR context**.
After generating the reasoning chain, please review it and
indicate your confidence in the reasoning chain at the end.
Options of confidence:  [Very Confident, Confident, Neutral, Not
Confident, Very Not Confident]
**Output Format:**
# Reasoning Chain #
1.  **Patient Overview**:
[YOUR OUTPUT]
2.  **Reasoning Towards Prediction**:
[YOUR OUTPUT]
3.  **Conclusion**:
[YOUR OUTPUT]
# Confidence #
[CONFIDENCE (choose one:  "Very Confident", "Confident",
"Neutral", "Not Confident", "Very Not Confident")]
```

---

Table 11: Prompt template for GPT-4o-based pairwise evaluation of reasoning quality. GPT-4o is instructed to compare the outputs of two models and select the one with more clinically grounded and coherent reasoning.

---

**Prompt Template (GPT-4o Evaluation)**

---

```
You are an expert in evaluating the quality of clinical
reasoning.  Your task is to assess the reasoning processes of
two models that predict whether a patient will receive their
first diagnosis of a specific condition within the next year,
based on the patient's historical clinical events.
Here are the inputs:
Clinical Events:
{formatted_text}
Timestamp:  {prediction_timestamp}
Ground Truth Label:  {target}
------------------
Two models gave the following reasoning and predictions:
Model A Reasoning and Prediction:
{model_a}
Model B Reasoning and Prediction:
{model_b}
Please evaluate which model provided a better reasoning process.
When evaluating the reasoning quality, consider the following
aspects:
- Relevance of Evidence:  The reasoning should be grounded in
clinically relevant information such as symptoms, medications,
laboratory results, and risk factors, with a clear and
purposeful selection of evidence.
- Causal and Temporal Reasoning:  The reasoning should reflect a
detailed and logically ordered understanding of the causal and
temporal relationships among clinical events.
- Clinical Plausibility:  The reasoning should be medically
sound, aligned with established clinical knowledge and practice.
- Explanation Quality:  The reasoning should be well-structured,
coherent, and clearly articulate the logical steps taken to
reach the conclusion.
Answer in JSON format:
<think>
[Your thinking process here]
</think>
<answer>
{ "better_model":  "A" or "B", "explanation":  "your reasoning"
}
</answer>
```

---

### F.3.4  ASSESSING REASONING QUALITY VIA GPT-4O EVALUATION

To assess the quality of the generated reasoning chains beyond accuracy, we conduct a pairwise comparison between EHRMIND-RLVR and EHRMIND-SFT-RLVR using GPT-4o as a judge. For each example, we present GPT-4o with the same clinical input and prediction timestamp, along with reasoning chains and final predictions produced by two models. GPT-4o is then asked to evaluate which model provided a superior clinical reasoning process based on four criteria: relevance of evidence, causal and temporal reasoning, clinical plausibility, and explanation quality.

To mitigate potential ordering bias, we perform this evaluation bidirectionally: for each test case, we generate two prompts with swapped model orders (i.e., A vs. B and B vs. A). We then aggregate the results across both directions to determine a final outcome. Specifically, if GPT-4o favors `SFT-RFT` in more comparisons than `RFT`, we record a win for `SFT-RFT`; if the reverse holds, it's a win for `RFT`; otherwise, the outcome is marked as a tie. This pairwise evaluation strategy ensures a more

robust and unbiased comparison of reasoning quality between models. The prompt can be found in Table 11.

