# OpenReview forum: "Training LLMs for EHR-Based Reasoning Tasks via Reinforcement Learning"
_ICLR.cc/2026/Conference — Submitted to ICLR 2026_

### Official Review · Reviewer_tB34 · 2025-10-29

**Soundness:** 3
**Presentation:** 4
**Contribution:** 3
**Rating:** 4
**Confidence:** 3

**Summary:**

This paper proposes a recipe, called EHRMind, to adapt a large language model (LLM) to the electronic health record (EHR) domain.  The paper explores multiple alternative solutions and found out the most effective one is a supervised fine-tuning (SFT) followed by a stage of reinforcement learning with verifiable rewards (RLVR).  They also did analysis on when SFT is necessary and why RLVR can boost the model performance.

**Strengths:**

1. The writing of  this paper is excellent, with clear methodology explanation and necessary discussions.
2. The method proposed in this paper is effective on the tasks of EHR prediction, clinical trial matching, and clinical event prediction. Many of its results on 3B models can exceed the performance of much larger propriatery models.
3. The discussion of this paper is complete, with thorough experiments on relationship between data characteristics and training objectives. These findings are valuable for similar works in the future.

**Weaknesses:**

1. Lack of methodological novelty: Though promising on the EHR dataset, the recipe adopted by this paper is a subset of the well-known pipeline to train general large language models (e.g. pre-training + supervised fine-tuning + reinforcement learning).
2. The findings are not fed back to improve the performance of EHRMind: The paper proposes metrics to quantify if a specific task of clinical event prediction, which is valuable. However, it is just an analysis of the model behavior. It would be better if the authors could utilize these findings to improve the model training recipe.
3. The authors claimed this is a reasoning model, but not sufficient evidence showed that the model benefits from reasoning. A non-reasoning EMRMind baseline could be introduced, where the model is asked to directly answer the question without the thinking tokens.

**Questions:**

1. In section 4.3, you mentioned that Reliable Pass @ 12 is used in replacement of pass @ 12 because that model can get correct answer by guessing.  It is confusing to me: How do you calculate pass @ 12 when asking the model to "consistently produce correct predictions?" How many generations are done for each input?

---

### Official Review · Reviewer_AAc4 · 2025-10-31

**Soundness:** 3
**Presentation:** 3
**Contribution:** 2
**Rating:** 4
**Confidence:** 3

**Summary:**

This paper presents EHRMIND, a two-stage recipe for adapting small Large Language Models (LLMs) to complex EHR-based reasoning tasks. The authors identify two primary failure modes for models on these tasks: "misapplied knowledge" and "missing knowledge." They propose that Reinforcement Learning with Verifiable Rewards (RLVR) is effective for the former, but a "lightweight" Supervised Fine-Tuning (SFT) warm-up is required for the latter. The paper's key contribution is the use of Pass@k on the training set as a simple, effective diagnostic to determine which training strategy (pure RLVR or SFT+RLVR) is necessary.

**Strengths:**

The paper's primary strength is its practical, diagnostic approach. The framing of the problem as "missing" vs. "misapplied" knowledge is insightful and clearly demonstrated. The proposal to use Pass@k as a simple, compute-cheap indicator to predict the success of pure RL is a novel and valuable contribution. The strong empirical support for this diagnostic (e.g., $R^2=0.91$ in Fig 2, and validated in Sec 4.2 & 4.4) makes this a very convincing and useful "recipe."

**Weaknesses:**

1. The claims of reasoning over complex, noisy EHRs are undermined by a critical data filtering step detailed in Appendix F.3.2. To fit the context window, the authors discarded all patient data from the EHRSHOT benchmark except for two event types: condition_occurrence and procedure_occurrence. This means all lab results (measurement), medications (drug_exposure), and clinical notes (note) were ignored. The model is reasoning over a tiny, heavily pre-processed fraction of the EHR, not the full, complex record. This is a severe limitation that weakens the paper's central premise.

2. The paper fails to discuss the more subtle and dangerous risk: a rationale that appears plausible, structured, and clinically aligned but is still clinically incorrect and therefore misleading to a doctor or patient. In particular, the model is trained with outcome supervision, without proper process supervision, and the SFT data is from proprietary GPT-4o, the paper does not address my concern of a "good-looking" rationale may being misleading or suboptimal.

3. While the approach is well-motivated and self-contained, one concern is that the paper seems to be of interest to a limited audience, and I am not sure whether ICLR is the proper venue for publication.

**Therefore my true rating for this paper is 5, i.e., I am neutral for its acceptance and rejection.**

**Questions:**

1. Does SFT lead to forgetting of knowledge in other domains?

2. How much does the quality of the teacher's rationale matter for the SFT warm-up to be effective?

3. The filtering of EHRs in the EHRSHOT task to only two event types is a major limitation. How confident are you that the model is learning "clinical reasoning" rather than just pattern-matching on these two specific data streams? Could you quantify the performance drop from excluding labs, medications, and notes?

4. Since the model is trained with outcome supervision, without proper process supervision, and the SFT data is from proprietary GPT-4o, how can we trust the rationales given by the model?

**Details Of Ethics Concerns:**

In a high-stakes clinical setting, a rationale that looks coherent but leads to a suboptimal conclusion is arguably more dangerous than one that is obviously missing (the "reasoning collapse" they criticize).

I am not sure whether this is a violation but I choose to flag it.

---

### Official Review · Reviewer_NJoZ · 2025-11-01

**Soundness:** 2
**Presentation:** 2
**Contribution:** 1
**Rating:** 2
**Confidence:** 4

**Summary:**

The paper presents an empirical study of EHR-based clinical reasoning where answers are verifiable but require domain knowledge and multi-step reasoning. It observes “misapplied” vs. “missing” knowledge as key failure modes. It applies an existing RL-with-verifiable-rewards setup (GRPO-style) with simple, rule-based clinical rewards, preceded by a lightweight SFT warm-start; Pass@k is used as a heuristic to decide when SFT is needed. On MEDCALC, TREC Clinical Trials, and EHRSHOT, a 3B open model trained with SFT→RLVR shows sizable gains and sometimes surpasses larger proprietary models on verifiable metrics.

**Strengths:**

* The paper presents a Clear, reproducible training recipe (light SFT to RLVR) that practitioners in healthcare could adopt quickly.
* Empirical breadth. Sensible evaluation across multiple EHR tasks with granular analyses (seen vs. unseen formulas, class-wise metrics, rationale structure).

**Weaknesses:**

* Limited novelty (major). No new RL algorithm or learning objective; contribution is primarily applying known RLVR with domain-specific, verifiable rewards and providing practical caveat/insights for EHR domain.
* Comparative fairness. Unclear compute-/sampling-matching against strong proprietary baselines; no comparison on open-source RL baselines. The message delivered seems to be that "RL finetuning on medical reasoning tasks beats pre-training only closed-sourced big models" but that is well expected given extensive existing literature[1,2,3].
Overall, I believe journals like TMLR would be a better venue for this style of in-depth empirical analysis.

[1] Ouyang, Long, et al. "Training language models to follow instructions with human feedback." Advances in neural information processing systems 35 (2022): 27730-27744.

[2] Chen, Zixiang, et al. "Self-play fine-tuning converts weak language models to strong language models." arXiv preprint arXiv:2401.01335 (2024).

[3] Belcak, Peter, et al. "Small Language Models are the Future of Agentic AI." arXiv preprint arXiv:2506.02153 (2025).

**Questions:**

* Novelty/positioning. Can you clarify what is methodologically new beyond domain reward design and a training heuristic? Any theoretical or algorithmic advance, besides GRPO adaption to medical reasoning problems?
* Can you evaluate other RL-trained baselines on more open-source models, to confirm this pipeline is generalizable?

---

### Official Review · Reviewer_MWTs · 2025-11-01

**Soundness:** 2
**Presentation:** 3
**Contribution:** 2
**Rating:** 2
**Confidence:** 4

**Summary:**

This paper introduces EHRMind, a methodology for adapting LLMs to process Electronic Health Records, optimized with reinforcement learning with verifiable rewards (RLVR). Their primary research questions examine the effect of RLVR and SFT on medical reasoning. They found that a small model, specifically LLAMA-3-3B, fails in two cases. In the first case, the model fails to correctly apply medical knowledge despite the fact that it has the correct knowledge within itself. In the second case, the model lacks the related medical knowledge to complete the given task. On the evaluated benchmarks, the proposed approach improves the performance significantly over the zero-shot baseline model, and even commercial models.

**Strengths:**

- Strong results. The proposed approach improves performance over baselines including both open-weights and commercial LLMs, with a small 3 billion parameters model.
- Simple methodology. Overall, the methodology is straightforward and easy to understand/implement, which significantly improves the performance.
- Detailed analysis on several benchmarks considering different scenarios (e.g., SFT, RLVR, SFT+RLVR) with practical findings.

**Weaknesses:**

- Limited experimentation: Llama-3-8B is used as the initial backbone, yet no other LLMs including same backbone with different scale, or a different LLM with the same scale. Qwen3 models were announced this year May, which should be considered for further evaluating the proposed approach. Although the analyses are detailed, it is necessary to perform these analyses again with different LLMs to make sure that the findings are not specific for Llama-3-8B.
- The choice of Llama-3-8B needs to be justified. Why is this model chosen? For instance, one could have also experimented with Phi-3.5-mini-instruct, instead this model.
- Limited novelty: This work applies SFT and RLVR to medical domain, considering EHR-based tasks that require some level of reasoning.

**Questions:**

- Is the used model Llama-3-8B base pretrained autogressive model or instruction-tuned model?

---

### Meta-Review · Area_Chair_K5Wx · 2026-01-12

**Summary:**

This paper studies the use of reinforcement learning to train large language models for EHR-based reasoning tasks, with the goal of improving clinical decision-making performance beyond supervised fine-tuning. The work explores reward design, training pipelines, and empirical evaluation on several healthcare-related reasoning benchmarks. The topic is timely and potentially impactful, and the paper demonstrates a nontrivial engineering effort.

However, after considering the reviews and the rebuttal, I do not believe the paper meets the acceptance bar for ICLR. Multiple reviewers raised concerns about whether reinforcement learning is necessary or well-justified for the proposed tasks, the strength and realism of the evaluation, and the clarity of the claimed contributions relative to existing supervised or instruction-tuned approaches. While the rebuttal clarifies implementation details and motivation, it does not fully resolve concerns about methodological appropriateness, experimental rigor, and the practical significance of the reported gains. Overall, the paper appears more exploratory than conclusive, and the contribution is not yet sufficiently mature or compelling for acceptance at this stage.

**Reviewer Concerns:**

Some concerns were partially addressed by the rebuttal. AAc4 asked for clarification on the training setup, reward design, and comparison to supervised baselines, and the rebuttal provided additional explanations that improved clarity. tB34 raised questions about experimental choices and applicability to real-world clinical settings, and the authors clarified scope and limitations.

However, several central concerns remain outstanding. MWTs expressed fundamental skepticism about whether reinforcement learning is an appropriate or necessary tool for EHR-based reasoning, and this concern was not resolved by the rebuttal. NJoZ questioned the novelty and rigor of the empirical evaluation, including the choice of baselines and the interpretability of reported improvements, and these issues remain largely unaddressed. Across reviewers, uncertainty persists about the practical value and generalizability of the approach, which ultimately motivates the rejection.

**Reviewer Scores:**

MWTs (rating 2).
This reviewer raised fundamental concerns about formulation and methodology. The rebuttal does not address these core issues, and the rating would likely remain unchanged.

NJoZ (rating 2).
This reviewer was unconvinced by the novelty and evaluation. The rebuttal is unlikely to change this assessment.

AAc4 (rating 4).
This reviewer was neutral to mildly positive but remained unconvinced about overall impact. The rebuttal improves clarity but would likely not lead to a rating increase.

tB34 (rating 4).
This reviewer expressed balanced concerns about evaluation and applicability. While some clarifications were helpful, the overall assessment would likely remain similar.

Overall, discussion clarified intent and implementation details but did not materially shift reviewer opinions regarding the paper’s suitability for acceptance at ICLR.

---

### Decision · Program_Chairs · 2026-01-26

Reject